# “*Sfogliatella Riccia Napoletana*”: Realization of a Lard-Free and Palm Oil-Free Pastry

**DOI:** 10.3390/foods10061393

**Published:** 2021-06-16

**Authors:** Raffaele Romano, Alessandra Aiello, Lucia De Luca, Alessandro Acunzo, Immacolata Montefusco, Fabiana Pizzolongo

**Affiliations:** 1Department of Agricultural Sciences, University of Naples Federico II, Via Università 100, 80055 Portici, Italy; rafroman@unina.it (R.R.); alessandra.aiello@unina.it (A.A.); lucia.deluca@unina.it (L.D.L.); montefusco.imma@libero.it (I.M.); 2Foodtech s.r.l., Galleria Vanvitelli 33, 80133 Naples, Italy; direzione@foodtechsrl.it

**Keywords:** vegetable fat, shea butter, sunflower oil, coconut oil, peroxides, total polar compounds, fatty acids

## Abstract

“*Sfogliatella riccia napoletana*” is a typical pastry from Naples (Italy), traditionally produced using lard. In the bakery industry, palm oil is widely used to replace lard in order to obtain products without cholesterol, but it is currently under discussion, which is mostly related to the sustainability of its cultivation. Therefore, in this work, lard was replaced with palm oil-free vegetable blends composed of sunflower oil, shea butter, and coconut oil in different percentages. Traditional pastries produced with lard and pastries produced with palm oil were used as controls. Moisture, a_w_, free acidity, peroxide value, fatty acids, total polar compounds, and global acceptability were determined in the obtained pastries. The results indicated that the use of a vegetable oil blend composed of 40% sunflower oil, 40% shea butter, and 20% coconut oil minimized the formation of oxidized compounds (peroxides and total polar compounds) during cooking and produced a product with a moisture content very similar to that of the traditional pastry that was appreciated by consumers.

## 1. Introduction

Many consumers are becoming increasingly interested in rediscovering local traditions. “*Sfogliatella riccia napoletana*” (SRN) is a lobster-tail-shaped filled Italian pastry native to Campania, in particular in the Neapolitan area, that is composed of layers of thin puff pastry overlapping each other and separated (laminated) by alternate fat layers, with a flaky texture. SRN is a type of “s*fogliatella*”, a pastry recognized as a traditional agri-food product of the Campania district and is placed in the category of bakery products and pastry according to G.U. 2001 [1]. The two types of “s*fogliatella*” pastry are known as “s*fogliatella riccia*” and “s*fogliatella frolla”*, a less labour-intensive pastry made with a short crust dough and without thin puff pastry layers.

The production of SRN is a complicated process that involves traditional production practices, and its production remains an art. In the classical process, flour, water, sugar, and salt are mixed in one stage to obtain the base dough. After mixing, the dough is divided into pieces and continuously sheeted and folded in a process commonly called “laminating” or “layering” the pastry, and the dough is extruded into a dough sheet. Subsequentially, the laminating fat, usually lard, is deposited as a continuous layer of fat on top of a continuous layer of dough. The fat is melted, applied to the dough sheet by a shortening extruder and absorbed by the dough, giving it a soft texture and resistance to moisture gain, which helps retain crispness when wet fillings are used in the puff pastry. Finally, the pastry is rolled into a log (much like a Swiss roll but with many more layers) and is allowed to rest. During the laminating process, dough layers are reduced to an average thickness of approximately 2–3 mm. Disks are then cut from the end, shaped to form pockets, and filled. Fillings include sugared ricotta, cooked semolina, eggs, candied orange cubes, and other flavourings. The pastry is baked in an oven until the layers separate, forming the SRN’s characteristic ridges.

Fats play a key role in bakery products, providing the desired rheological properties and specific sensory properties (aroma, flavour, softness, volume, palatability, bright appearance) as well as stabilizing the products against oxidation reactions, staleness, and moisture migration [2]. In particular, fats used in laminated products influence the texture and colour of the final product [3] and need to be in a solid or semisolid state at room temperature to facilitate the handling of the batter during manufacturing, which implies an increase in the content of saturated fatty acids [4]. Lard is the fat traditionally used in s*fogliatella* production and is responsible for its appreciated flaky structure. It is heat-rendered from the adipose tissues of pork (*Sus scrofa*) and subsequently refined. Typically, lard contains high levels of saturated fatty acids (~45%) and cholesterol (~97 mg/100 g) (which contribute to the development of cardiovascular disease), even if it shows low oxidative stability due to the absence of antioxidants [5]. There is a growing negative perception regarding the implication of animal fats on human health; thus, a significant trend towards the utilization of vegetable-derived shortening as a substitute for lard has been revealed. Many reasons are attributed to this trend, including vegetable oil availability, oil processing capability, nutritional needs, cost, bulk handling, storage, and labelling [6], but studies addressing the formulation of fat substitutes to mimic the physical characteristics of lard are still limited [7]. Towards the end of the nineteenth century, developments in refining technology and fat modification techniques allowed the use of a widening range of fats, such as coconut oil, soybean oil, palm oil and palm kernel oil, as less expensive and zero-cholesterol substitutes for animal fats and in margarine and shortening production. Currently, vegetable oils are of great interest, but most of them have to be transformed to reach the necessary functionality in such products. Partial hydrogenation, interesterification, fractionation and bleaching, deodorization, winterization, and refining are common fat modification processes [8,9]. It is well-known that partial hydrogenation led to the formation of trans fatty acids that may have adverse consequences on human health [8]. Sunflower, palm, soybean, and coconut are the major raw materials for the production of vegetable oils around the world, although the usefulness of indigenous and semidomesticated plants and trees as vegetable oil sources, such as shea butter obtained from the seeds of *Vitellaria paradoxa*, is starting to percolate into the awareness and knowledge of agro-foresters, entrepreneurs, and the public [10,11,12,13]. Vegetable oils can be used alone or in blends. Several studies show that the formulation of blends is a key factor affecting final product quality, but processing is also a critical point [14,15,16]. Blending oils with different levels of unsaturation can improve the functional, nutritional, and technical attributes of the oil and is a desirable alternative to the negative effects associated with hydrogenation [15].

Palm oil is currently under discussion, which is mostly related to the sustainability of its cultivation. Approximately 185 million tons of palm oil are produced in the tropics and are then traded globally. Unfortunately, the highest concentrations of vegetable and animal species (and the highest biodiversity) are also in the tropics; thus, the doubling of palm oil production from 2003 to 2013 has raised concerns about possible wildlife extinction [17]. Deforestations have occurred, and farmers have cleared forests quite recently. In these countries, palm plantations threaten iconic and endangered wildlife, including orangutans, Bornean pygmy elephants, and the Sumatran tiger. Similarly, palm plantations in South America are also the products of recent deforestation. Moreover, tropical deforestation also releases the carbon stored in tree tissues and in the soil, contributing to an estimated 10% of the greenhouse gas emissions that cause global warming. The demand for palm oil is increasing, so there is concern that palm plantations will spread, resulting in the deforestation of other parts of the world, such as Indonesia, Malaysia, Papua New Guinea, and Brazil (particularly in the Amazon), where animal species such as the Little Woodstar, the largest species of armadillo, and numerous brightly coloured poison frogs could be threatened. In Indonesia, Malaysia, Ecuador, and Peru, most of the plantations have been deforested since 1990, and oil palm was likely the cause of much of this deforestation [17]. With the prospect of continued high demand in the largest importing countries (India, China, and Europe), strategies for sustainable palm oil production represent a necessary step forward. There also seems to be an emerging trend in palm oil alternatives in European countries, with the use of vegetable blends that have the same technological properties.

In this work, the replacement of lard traditionally used in the formulation of SRN with a mixture of palm oil-free vegetable fats having the same technological properties as lard was studied to obtain SRN without cholesterol and with low trans fatty acid content.

## 2. Materials and Methods

### 2.1. Fats

Lard (L) was used to produce the control SRN. Palm oil, sunflower oil, coconut oil, and shea butter blended in different proportions to obtain the same melting characteristics of lard were used in experimental SRN formulations. All fats were purchased from the local market. Fat blends were obtained by an enzymatic interesterification process so that they had levels of saturation similar to lard (39.4%) and were coded as follows: P, palm oil 80% and sunflower oil 20%; V1, vegetable fat blend, composed of sunflower oil 60%, shea butter 25%, and coconut oil 15%; V2, vegetable fat blend, composed of sunflower oil 40%, shea butter 40%, and coconut oil 20% (Table 1). These blends were stored at −20 °C in a closed container until their use in SRN production and until the following analyses were conducted: free acidity or free fatty acids (FFA), peroxide value (PV), fatty acids (FA) composition, and triacylglycerols (TAG).

#### 2.1.1. Free Acidity

Free acidity is the amount of FFA, expressed as g of oleic acid, present in 100 g of oil [18]. Acidity is a measure of the hydrolysis of the oil’s triglycerides: as the oil degrades, more fatty acids are freed from the glycerides, increasing the level of free acidity, thereby increasing the hydrolytic rancidity. The acidity was determined according to the European official methods of analysis [19] by titration of a solution of oil dissolved in ethanol–ether (1:2) with sodium hydroxide.

#### 2.1.2. Peroxide Value

Another measure of fat chemical degradation is the peroxide value [20], which measures the formation of intermediate hydroperoxides in milliequivalents of active oxygen per kilogram of oil (meq O_2_/kg). The peroxide value of fat was determined according to annex III in Reg. CE n. 1989/2003) [21]. An aliquot of fat (1.5 g) was transferred into a dry 125-mL Erlenmeyer flask, and 15 mL of acetic acid and chloroform solution (3:2) were added. The flask was swirled, and 0.25 mL of saturated potassium iodide solution was added. After allowing the solution to remain in a dark place for 10 min, 15 mL of distilled water was added. This solution was titrated against 0.1 N sodium thiosulfate after the addition of 1 mL of starch indicator (1% *w*/*w*) until the blue colour of the solution had disappeared.

#### 2.1.3. Fatty Acids Composition

FA composition analysis was performed by gas chromatography (GC) following derivatization to FAMEs with 2N KOH in methanol according to IUPAC standard method n^o^. 2.301 [22]. First, 50 mg of fat was dissolved in 1 mL of n-hexane in a 1-mL glass-stoppered test tube. A solution of 2N potassium hydroxide in methanol (300 μL) was added. The tube was vortexed for 30 s and allowed to react for 4 min at room temperature. A 1-μL aliquot of the upper organic phase was analysed using high-resolution gas chromatography (HRGC). A DANI Master gas chromatograph (Dani Instrument SPA, Milan, Italy) equipped with a programmed temperature vaporizer (PTV), a flame ionization detector (FID), and an SP-2380 fused silica capillary column (Supelco Inc., Bellofonte, PA, USA) with the dimensions of 100 m × 0.25 mm i.d. and a film thickness of 0.20 μm were used, as previously described by Romano et al. [23]. The oven temperature was programmed to hold at 80 °C for 5 min, followed by a 5 °C/min ramp-up to 165 °C for 5 min, and then a 3 °C/min ramp-up to 230 °C for 1 min. The carrier gas, helium, was introduced at a flow rate of 20 cm/s. An aliquot of 1 μL was injected into the injection port, with a split ratio of 1/60 (*v*/*v*), and the FID temperature was set at 260 °C, with an 8:1 ratio of air:hydrogen. The PTV operating conditions were held at 50 °C for 0.1 min, after which the temperature was increased at 400 °C/min up to 230 °C and was then held for 1 min. The identification of the peaks was performed using an external standard (SupelcoTM 37 component FAME MIX). The sample concentrations were calculated through a comparison with the pure standard retention time and were based on response factors to convert peak areas into weight percentages (mg/100 g of fatty acids), as suggested by Molkentin [24].

#### 2.1.4. Triacylglycerols and Cholesterol 

One microlitre of a solution of 4% (*w*/*w*) fat in hexane was injected into the gas chromatograph. The TAG solution was analysed on a GC Dani 1000 gas chromatograph equipped with a flame ionization detector (FID) held at 360 °C and a temperature-programmed vaporizer (PTV) injector. An Rtx 65TG (mod. 17008) capillary column (30 m × 0.25 mm i.d.; 0.10 μm film thickness) from Restek (Bellefonte, PA, USA) was used with high-purity helium as a gas carrier at 1.5 mL/min. The injection was performed with a split ratio of 1/30 (*v*/*v*) and a heating programme of 50 °C for 0.3 min, followed by a 500 °C/min increase to 300 °C with a 7-min hold. The column oven temperature programme was 250 °C for 2 min, followed by an increase of 6 °C/min to 360 °C for 10 min, as described in Romano et al. [25]. The flow rate of the gas carrier, He, was 1.2 mL/min. TAGs were identified by comparing their retention times with those of a triglyceride standard (Sigma, Saint louis, MO, USA). These samples allowed response factors (Rfs) to be calculated, which converted peak areas into weight percentages. The reference standard for cholesterol was purchased from Sigma Chemical Co. (Saint louis, MO, USA).

### 2.2. SRN Formulation

SRNs were produced at a confectionery company in Naples according to the traditional process (Figure 1). The puff pastry was prepared using water (11%), salt (1%), sugar (1%), and wheat flour (33%). The wheat flour (15.4% protein, 14.0% moisture, and 0.65% ash) was supplied by Molino Sul Clitunno s.r.l (Trevi, Italy). These ingredients were mixed with a fork mixer (INM 200, Pietroberto s.r.l, Marano Vicentino, Italy) at 22 rpm for 15–20 min. The short pastry was laminated to obtain pastry circles 9 cm in diameter, and melted fat (8% total fat) was added. The filling was obtained by mixing cooked semolina (23%), sugared ricotta (19.5%), pasteurized eggs (1.8%), candied orange cubes (1.5%), and cinnamon, vanilla, and orange (used as flavourings) (<1%) in a spiral mixer and was then mixed until a homogeneous mix was obtained. SRNs with a weight of approximately 110 g were produced. Traditional SRN was produced by adding fat L to layers and was used as a control. In the experimental formulations, lard was replaced with the same weight of fat blends (P, V1, and V2) according to the same process used in the SRN control, keeping the ratios among the other ingredients constant. The SRN samples were coded SRN-L, SRN-P, SRN-V1, and SRN-V2. SRNs were baked in a commercial electric oven (Electrolux Rex, Porcia, Italy) at 200 °C for 45 min. The oven was preheated to the set temperature before placing the samples into it. SRNs were cooled at room temperature (20–22 °C) for 1 h and stored at −20 °C in sealed plastic containers until analysis. An aliquot of the SRNs before baking was also stored under the same conditions.

The SRNs were submitted to the following analyses: moisture content, water activity, fat content, FFA, PV, FA composition, total polar compounds (TPC), and sensory characteristics. Moisture and water activity were determined in the baked SRNs in both the filling and the puff pastry. Fat content, FFA, PV, FA, and TPC were determined in the puff pastry of both raw and baked SRNs. Sensory analyses were performed immediately after baking.

#### 2.2.1. Moisture Content

The moisture content was determined by the gravimetric method [26] in the puff pastry and filling samples. Samples were oven-dried at 105 °C and were accurately weighed at regular time intervals until a constant weight was reached. Three measurements were performed for each sample, and the average was determined. The moisture content was expressed as grams of water over grams of total weight (g/100 g): $$\frac{\left(b-a\right)}{p}\times 100$$
in which:*a* = weight of cap with dried sample;*b* = weight of cap with not-dried sample;*p* = weight of sample in grams.

#### 2.2.2. Water Activity (a_w_)

Water activity (a_w_) was measured using an AquaLab Series 4TE water activity meter (Decagon Devices Inc, Pullman, WA, USA) in AwE mode. The a_w_ meter was calibrated using verification standards (a_w_ 0.984 and a_w_ 0.760) from Decagon (NE Hopkins Ct. Pullman, WA, USA) at 25 °C. For each of the five water activity levels, the values were determined within the required range of 0.003. Water activity was measured on both the homogenized puff pastry and the filling, and approximately 2 g was placed in a plastic box and introduced into the moisture sensor of the equipment. The a_w_ levels were measured in triplicate in 3 independent baking batches 2 h after baking at 25 °C.

#### 2.2.3. Fat Extraction

Extraction of the fat from the SRNs before (raw samples) and after baking (baked samples) was carried out according to the Schmid-Bondzynski-Ratzlaff method of lipid extraction [27] with some modifications. First, the samples (10 g) were homogenized with ethanol (7 mL) and mixed using a vortex mixer for 60 s. A diethyl ether-heptane mixture (10 mL, 2:1 *v*/*v*) was then added and mixed by vortexing for 60 s. Samples were then centrifuged at 8000 rpm for 10 min. The diethyl ether phase containing the extracted lipids was transferred, and the residue was extracted using the same procedure three more times. The combined filtrates were concentrated in a rotary evaporator at 36 °C. The extracted lipid phase was then dissolved in hexane and purified using sodium chloride saturated solution (3 mL). The hexane phase containing purified lipids was dried over anhydrous sodium sulfate and under nitrogen. The total lipids obtained were determined gravimetrically using the AOAC method [28]. The extracted fat was stored in glass tubes at −20 ± 0.2 °C until the analyses.

#### 2.2.4. Free Acidity, Peroxide Value and Fatty Acids

FFAs, PV, and FAs were determined in the raw and baked SRNs as previously described for fat used as raw material.

#### 2.2.5. Total Polar Compounds

TPCs were determined using silica minicolumn chromatography according to the method indicated by Dobarganes et al. [29] based on the selective adsorption of polar and non-polar lipids on a silica gel column, followed by non-polar compound elution with the appropriate solvents. Subsequently, both fractions were gravimetrically quantified. The polar compounds include polar substances such as monoacylglycerols, diacylglycerols, and free fatty acids, which occur in unused fats, as well as polar transformation products formed during the frying of foodstuffs and/or during heating. Non-polar compounds are mostly unaltered triacylglycerols. Petroleum ether/ethyl ether (ratio 90/10) was used to elute the non-polar fraction (non-altered triglycerides) to obtain a sharper separation, and a final elution of the column was made with CH_3_OH to improve the recovery of the sample. Polar compounds were evaluated in triplicate using a silica minicolumn (Discovery SPE DSC-Si silica tube, 20 mL, 5 g; Supelco Analytical). The results are expressed as percentages (*w*/*w*) by the following formulas: $$wNP=\frac{mNP}{m}\times 100\;\;\;w=(\frac{mp}{m})\times 100$$
where:*mNP* = weight in grams of nonpolar fraction*mp* = weight in grams of polar fraction*m* = weight in grams of sample

#### 2.2.6. Sensory Evaluation

To test the overall acceptability of the product, sensory analysis was conducted using a semi-structured hedonic scale [30,31]. A 42-member panel of untrained consumers with some experience in the sensory evaluation of SRN was recruited. The panellists were 18 women between 17 and 60 years old and 24 men between 19 and 64 years old. The SRN samples formulated with different fat blends (L, P, V1, and V2) were baked, labelled with a random 3-digit code and served individually to the panellists in plastic cups in random order. All samples were presented in duplicate with different sample orders. The tests were performed in an isolated room with good illumination and natural ventilation in groups of 7 subjects at a time. Panellists rinsed their mouths with still water between samples. Global acceptability was evaluated to study the effects of the different fat formulations. Each panellist received a form sheet with a nine-point hedonic scale anchored with “Like Extremely” and “Dislike Extremely” at either end, with a neutral point of “Neither Like nor Dislike”.

### 2.3. Statistical Analysis

All analytical determinations were repeated three times for each preparation method and the reported results represent the averages of the values obtained (±standard deviations). One-way analysis of variance (ANOVA) and Tukey’s multiple-range test (*p* ≤ 0.05) were conducted on the data using XLSTAT software (Addinsoft, New York, NY, USA).

## 3. Results and Discussion

### 3.1. FFA and PV in Fat Blends

Data regarding the free fatty acids (FFAs) and peroxide values (PVs) of the fat blends used in the production of SRN are shown in Figure 2. Lipids are susceptible to hydrolytic and oxidative processes. These processes can be accelerated if the fat is exposed to high temperatures, such as during baking, causing thermal oxidation with increases in free fatty acid and polar matter contents [32]. FFAs and PV are among the most common quality indicators of fats and oils during production and storage. Regarding FFAs, no statistically significant differences (*p* < 0.05) emerged between the lard and vegetable fat blends. PV is a marker of the initial stages of oxidative change. By studying the kinetics of hydroperoxide concentration, it is possible to assess whether a lipid is in the growth or decay portion of the reaction process [32]. Calligaris et al. [33] reported that PV could be considered a good chemical index to monitor the loss of the sensory quality of biscuits during their shelf life, as it was well correlated with sensory consumer acceptance. As expected, lard (control) showed the highest PV values. Lard is a semisolid fat commonly used in the food industry, especially in the preparation of bakery products, even if it contains practically no natural antioxidants [34], and oxidative degradation can occur. Thus, recent trends in the baking industry are aimed at replacing lard and other animal fats with vegetable fats [7].

### 3.2. FA composition of Fat Blends

The mean values and standard deviations of the fatty acid composition of the fat blends utilized in SRN-making are shown in Table 2. The saturated fatty acid (SFA) contents were similar in V1 and L (40.35 ± 0.31% and 39.4 ± 0.19%, respectively) and in V2 and P (42.43 ± 0.23% and 42.97 ± 0.28%, respectively). The content of the SFAs positively influenced the plastic properties of fat blends [35]. However, the fat blends showed different fatty acid profiles. The main saturated fatty acid present in L and P was palmitic acid (C16:0), with values of 24.10 ± 0.11% and 35.31 ± 0.23%, respectively, while V1 and V2 showed high contents of stearic acid (C18:0) (20.37 ± 0.19% and 26.95 ± 0.03%, respectively). The high stearic acid content in V1 and V2 are attributed to the presence of shea butter, which is rich in this fatty acid [36]. Thus, these fat blends without palm oil make fats attractive on nutritional grounds since stearic acid contributes to reduced plasma LDL concentrations and, hence, have no adverse effect on the risk of cardiovascular disease [37]. The vegetable fat blends V1, V2, and P differed from L in terms of their higher poly-unsaturated fatty acid (PUFA) content due to the presence of sunflower oil (Table 2). The high contents of linoleic acid (C18:2n6c) in V1, V2, and P (37.44 ± 0.02%, 27.32 ± 0.06%, 19.57 ± 0.18%, respectively) correspond to different concentrations of sunflower oil incorporated into the blends as the liquid component. Vegetable fat V1 showed the highest value of PUFAs and the lowest value of MUFAs due to the high percentage of sunflower oil in the mixture (60%). L contained a high level of mono-unsaturated fatty acids (MUFAs) (44.47 ± 0.17%, with 16.13 ± 0.03% of PUFAs). The main MUFA was oleic acid (C18:1n9c) (41.33 ± 0.17%) as also reported by Chiesa et al. [38]. The MUFA/SFA, PUFA/SFA, and UFA/SFA ratios (Table 2) showed different fatty acid profiles. The UFA/SFA ratio was higher in L. The two vegetable fats without palm oil exhibited the lowest MUFA/SFA ratio and the highest PUFA/SFA ratio. Practically negligible contents (<1%) of trans fatty acids (TFAs), in accordance with the required values [39], were detected in L, V1, and V2 (0.26%, 0.18% and 0.25%, respectively). They were not detected in P.

### 3.3. Triacylglycerols and Cholesterol in Fat Blends

Fat blends consist of a mixture of a large number of TAGs, as shown in Table 3. To have fat in the ß‘ crystal form, the most stable form, the TAG carbon number should be kept as low as possible, preferably below C54 [40]. L mainly contains C52, which is ß‘ tending. The C52 content is reduced in V1 and V2. Moreover, the C54 content is reduced in L but increased in V1 and V2. Additionally, V1 and V2 contain higher contents of C24 to C42 due to the presence of coconut oil in the blend (Table 1). It has been reported that at 15 °C, margarines with palmitic acid (C16:0) below 11% are in ß form, while those with 50% or below are in ß′ crystal form. ß′ has relatively very small crystals, which enable it to incorporate relatively large amounts of liquid oil in the crystal network. This phenomenon leads to the production of smooth, continuous, and homogeneous products. Shortenings and margarines containing ß‘ crystals appear smooth and shiny in contrast to those containing ß crystals, which produce a dull and mottled product [40]. Because of the high content of palmitic acid (35.31 ± 0.23%), P tends to ß‘ and contains mainly C50 and C52 (32.0 ± 0.2% and 34.4 ± 0.4%, respectively). As expected, cholesterol content was only found in L (110 mg/100 g).

### 3.4. Moisture Content and Water Activity in SRN

As expected, the moisture content of the filling was higher than that of the puff pastry (Figure 3). The filling moisture content ranged between 45.4% and 54.0%; however, the puff pastry showed a moisture content between 10.9% and 15.2%. All vegetable SRNs (P, V1, and V2) exhibited moisture content lower than L in both the filling and puff pastry samples. SRN-P showed the lowest moisture content. Animal fat seemed to coat the layer surface more efficiently and limited moisture migration during baking. SRN-V1 exhibited a moisture content in the puff pastry similar to traditional SRN-L. Considering the moisture content in the filling samples, SRN-V2 exhibited a moisture content similar to that of the traditional sample.

Water activity (a_w_) describes the contribution of free water to the support of bacteria and fungi. Quantifying a_w_ is the key to research on food preservation and safety [41]. Similar to moisture, the water activity values of the filling (0.92–0.96) were higher than the values of the puff pastry (0.78–0.83), as reported in Figure 4. V2 exhibited the highest a_w_, while P and V1 showed values lower than L.

### 3.5. FFA, PV and TPC in SRN

The chemical characteristics of the raw and baked SRNs made with different fat blends (L, P, V1, and V2) are shown in Table 4. The formation of free fatty acids might be an important measure of the rancidity of foods. FFAs are formed due to the hydrolysis of triglycerides [42]. The acidity values ranged from 0.11% to 0.58% in raw SRNs. An appreciable change in acidity was observed in the samples after baking (*p* ≤ 0.05). The acidity of SRN-L was 0.58% in the raw product and 0.72% in the baked product. SRN-V1 showed values similar to the control sample (SRN-L), in particular, it exhibited the highest value of 0.83% after baking. V2 exhibited the lowest value of acidity in comparison with the other fats (0.11 and 0.18% before and after baking, respectively). The FFAs level depends on temperature, moisture content, and time [43].

The analysis of the peroxide value measures hydroperoxides, which are transient products of lipid oxidation. As lipid oxidation progresses, hydroperoxide formation increases, and the peroxide value also increases [44]. Raw samples exhibited PV between 2.88 and 3.53 meq O_2_/kg. The PV content increased after baking in all the samples, with L exhibiting the highest values (15.75 meq O_2_/kg) and V2 exhibiting the lowest (5.35 meq O_2_/kg). The results obtained for FA and PV were in line with DaĞlioĞlu et al. 2000 [45], who also reported an appreciable increase in the acidity and peroxide values in samples of puff pastries after baking.

The level of polar compounds is a good indicator of the quality of heated fats and oils and is the most common method of evaluating the alteration of fats due to its accuracy and reproducibility [46]. It provides information on the total amount of newly formed compounds with polarity higher than that of triacylglycerols. The TPC of heated fats provides the most reliable measure of the extent of oxidative degradation [47]. The results showed that the contents of TPC, FFAs, and PV increased after baking. The initial TPC contents of fats extracted from raw SRN-L, SRN-P, SRN-V1, and SRN-V2 were 8.84, 9.17, 9.65, and 9.19%, respectively, which were significantly increased to 14.55, 15.22, 16.44, and 13.39% after heating. The maximum TPC content was for V1 (16.44%), followed by P and L. V2 showed the minimum value (13.39%). In conclusion, SRN-L had a very poor quality of fat; instead, SRN-V2 showed better thermo-oxidative stability, with the lowest FFA, PV, and TPC values and the lowest increases in PV and TPC after baking.

### 3.6. FA Composition of SRN

The major fatty acids in the raw samples (Table 5) were oleic (C18:1n9c), palmitic (C16:0), stearic (C18:0), and linoleic (C18:2) acids, while the remaining fatty acids, except lauric (C12:0), myristic (C14:0), and palmitoleic (C16:1) acids were present in concentrations of approximately 1% or less. The ratio of saturated to unsaturated fatty acids (UFAs/SFAs) did not change significantly for different fats, but the saturated fatty acid (SFA), monounsaturated fatty acid (MUFA), and polyunsaturated fatty acid (PUFA) contents were different. The SFA contents ranged from 40.56 to 48.53% of the total fatty acids, but the profiles were different. In SRN-L, SRN-P, and SRN-V1, the major SFA was palmitic acid, while in SRN-V2, it was stearic acid. It has been reported that palmitic acid contents above 17% tend to be in the β’ form, which is desirable for plastic fats, and contents below 11% tend to be in the β form [48]. Thus, the fat crystals were in the stabilized β’form in all of the SRNs. The total unsaturated fatty acid (UFA) contents ranged from 51.47 to 59.44%, and the major proportion of them was composed of monounsaturated fatty acids (MUFAs). However, SRN-V1 and SRN-V2 contained fewer MUFAs and a higher percentage of linoleic acid than SRN-L and SRN-P. The majority of SRNs contained TFAs, but always less than 0.5%, ranging from 0.23 to 0.42% of the total fatty acids. The highest amount of TFA was detected in SRN-L (0.42%). The predominant trans isomer found in all samples was C18:1n9t monounsaturated fatty acid. DagĞlioĞlu et al. [49] also reported that trans monounsaturated fatty acids with 18 carbon atoms (C18:1n9t) are typically abundant in cereal-based foods. Among trans polyunsaturated fatty acids, C18:2n6t was detected at concentrations of much less than 0.1%. TFAs were lower in vegetable fats than in L. Significant differences in the total fat and TFA levels among the analysed samples can be attributed to the composition of the fat used in SRN production.

During baking, the total SFAs and the oleic acid content of the SRN samples as well as the ratio of saturated to unsaturated fatty acids did not change significantly (*p* < 0.05). While differences in singular FAs were observed in SRN-L, SRN-V1, and SRN-V2 before and after baking (Figure 5), no statistically significant differences emerged in the SRN-P sample. Regarding the control, SRN-L, changes occurred in unsaturated fatty acids, particularly in the percentage of linoleic acid (C18:2n6c). Its content decreased from 13.8% (in raw SRN-L) to 12.9%. Significant increases in C4:0 and C14:0 were also observed after heating (Figure 5). Reductions in linolenic acid (C18:3n3) in SRN-V1 and linoleic acid (C18:2n6c) in SRN-V2 (Figure 5) were shown, with increases in some saturated fatty acids (C10:0, C12:0, C14:0) in the latter. DagĞlioĞlu et al., 2000, [45] also indicated a significant decrease in linoleic and linolenic acid between the raw and baked samples of puff pastry. These reductions in unsaturated fatty acids result from oxidation that occurs during baking. Yoshida et al. [50], indicated that thermal oxidative deterioration causes increases in non-volatile compounds such as free fatty acids. SRN-V2 also showed the reduction of C18:0. This is the most abundant saturated fatty acid, showing a higher value (19.85%) in SRN-V2 than in the other samples and is in the sn-1 and 3 positions of tricylglycerols. Therefore, it might become the preferential target of the lipase. The results indicate a decrease in oxidative stability in baked SRNs. In SRN-L, the trans oleic acid content increased somewhat after baking, but the difference was insignificant (*p* < 0.05). Trans linolenic acid was not present at a detectable level in SRN-L.

### 3.7. Sensory Evaluation

The results of the sensorial evaluation of baked SRNs are shown in Figure 6. No significant differences emerged among SRNs produced with different fat formulations (*p* < 0.05). In general, the SRNs produced with vegetable fat blends were well accepted by the consumers, with scores of approximately 7, corresponding to “Like Very Much”–“Like Moderately”. SRN-P presented the lowest scores.

## 4. Conclusions

Different vegetable fat blends were used to produce SRN puff pastry without cholesterol. All vegetable fat SRNs had a *trans* fatty acid content less than 0.4%, and moisture comprised between 10 and 15%. The vegetable fat blend composed of 40% sunflower oil, 40% shea butter, and 20% coconut oil (V2) allowed us to obtain a product with a lower *trans* fatty acid content and a higher thermal-oxidative stability than traditional SRN. In fact, the SRN produced with this blend showed the lowest values of FFA, PV, and TPC both in the raw and baked products. Moreover, using a nine-point hedonic scale, liking scores indicated that the vegetable fat SRNs had score similar those to the traditional pastry. This research contributed knowledge of replacing animal fats with vegetable fats in pastries, and in particular, in SRN, on which no studies have been previously described.

## Figures and Tables

**Figure 1 foods-10-01393-f001:**
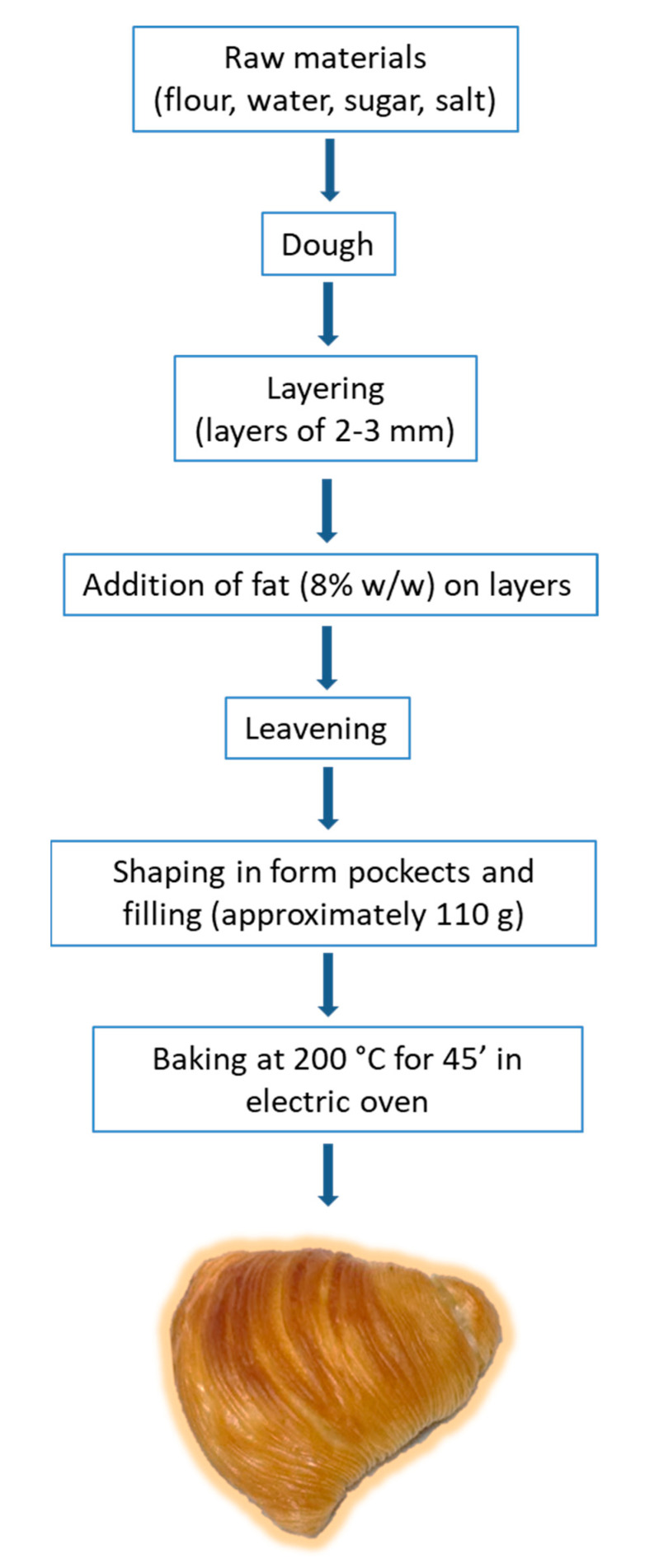
Production of “*sfogliatella riccia napoletana*” SRN.

**Figure 2 foods-10-01393-f002:**
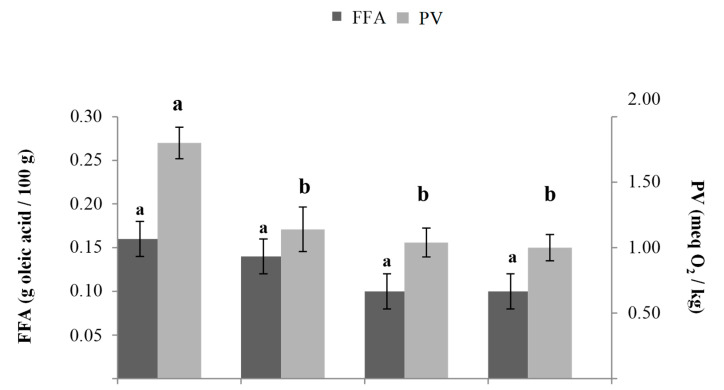
Free fatty acids (FFA) and peroxide values (PV) of the fat blends L, P, V1, and V2 (for identification of the samples see Table 1). a,b: Different letters correspond to statistically significant differences (*p* ≤ 0.05) for each parameter.

**Figure 3 foods-10-01393-f003:**
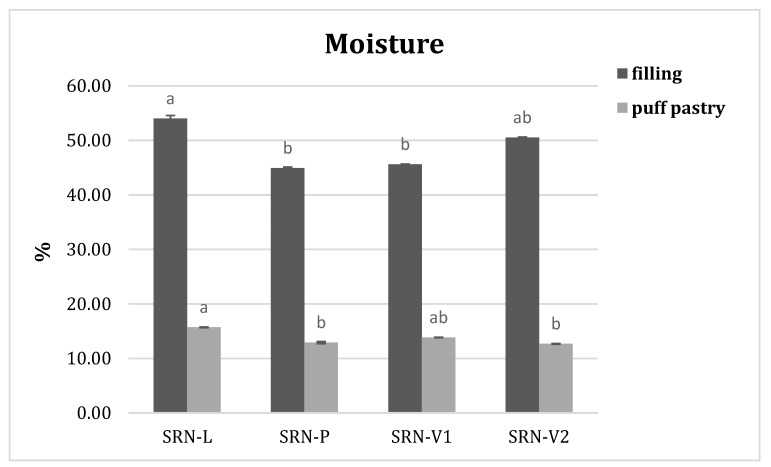
Moisture contents of the filling and puff pastry in baked “*sfogliatella riccia napoletana*” prepared with the different fat blends L, P, V1, and V2. a,b: Different letters indicate statistically significant differences (*p* ≤ 0.05).

**Figure 4 foods-10-01393-f004:**
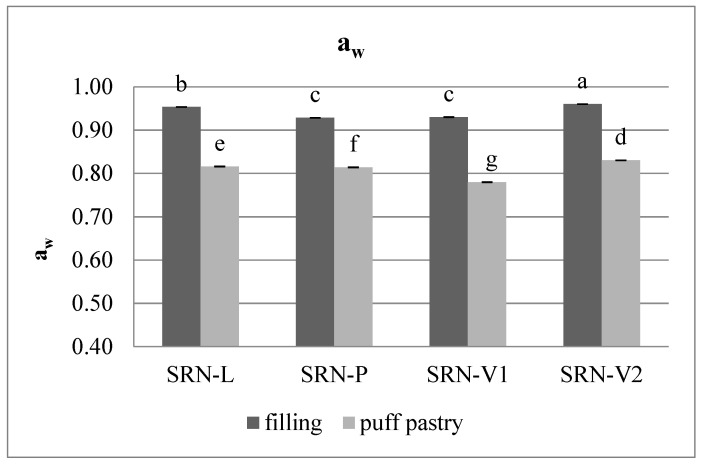
The a_w_ values of the filling and puff pastry in baked SRNs produced with the different fat blends L, P, V1, and V2. a–g: Different letters indicate statistically significant differences (*p* ≤ 0.05).

**Figure 5 foods-10-01393-f005:**
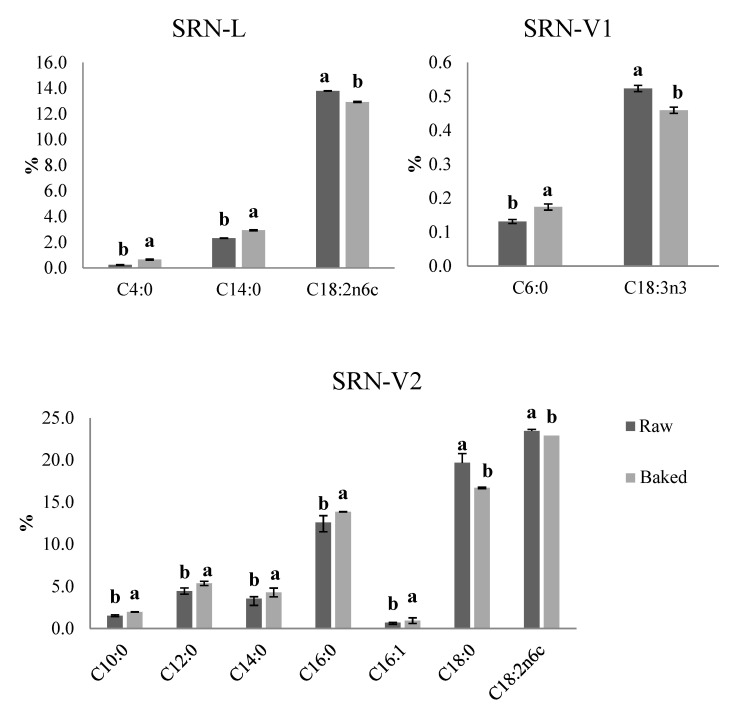
Changes in the fatty acid content of the raw and baked SRNs prepared with the different fat blends, L, V1, and V2. a,b: Different letters correspond to statistically significant differences (*p* ≤ 0.05) for each parameter.

**Figure 6 foods-10-01393-f006:**
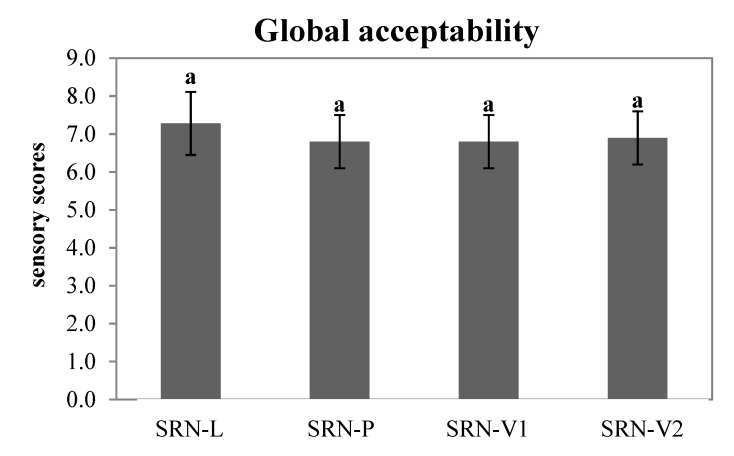
Global acceptability of the baked SRNs prepared with the different fat blends, L, P, V1, and V2. The same letters correspond to no statistically significant differences (*p* ≤ 0.05).

**Table 1 foods-10-01393-t001:** Compositions of fat blends used to prepare different “*sfogliatella riccia napoletana*” SRN.

Blend	Lard (%)	Palm Oil (%)	Sunflower Oil (%)	Coconut Oil (%)	Shea Butter (%)	Saturation (%)
L	100					39.4
P		80	20			42.9
V1			60	15	25	40.4
V2			40	20	40	42.4

**Table 2 foods-10-01393-t002:** Fatty acid compositions (mean ±SD) of the fat blends L, P, V1, and V2 (for identification of the samples see Table 1).

Fatty Acids (%)	L	P	V1	V2
C4:0	nd	nd	0.73 ± 0.64	0.43 ± 0.03
C6:0	nd	nd	0.07 ± 0.01	0.05 ± 0.01
C8:0	nd	nd	1.05 ± 0.03	0.74 ± 0.03
C10:0	0.07 ± 0.02	nd	0.81 ± 0.05	0.54 ± 0.01
C12:0	0.09 ± 0.01	0.21 ± 0.02	7.25 ± 0.06	4.76 ± 0.21
C14:0	1.50 ± 0.10	1.02 ± 0.03	2.72 ± 0.08	1.85 ± 0.01
C14:1	0.01 ± 0.01	nd	nd	nd
C15:0	0.04 ± 0.01	nd	nd	nd
C16:0	24.10 ± 0.11	35.31 ± 0.23	6.21 ± 0.23	5.87 ± 0.02
C16:1	2.63 ± 0.14	0.17 ± 0.01	0.03 ± 0.02	Nd
C17:0	0.27 ± 0.01	0.06 ± 0.01	0.04 ± 0.00	0.06 ± 0.01
C17:1	0.27 ± 0.01	nd	0.01 ± 0.01	0.01 ± 0.01
C18:0	13.14 ± 0.09	5.56 ± 0.05	20.37 ± 0.19	26.95 ± 0.01
C18:1n9t	0.23 ± 0.01	nd	0.09 ± 0.14	0.18 ± 0.01
C18:1n9c	41.33 ± 0.17	36.84 ± 0.02	21.73 ± 0.04	29.59 ± 0.03
C18:2n6t	0.03 ± 0.01	nd	0.09 ± 0.09	0.07 ± 0.00
C18:2n6c	13.85 ± 0.05	19.57 ± 0.18	37.44 ± 0.02	27.32 ± 0.06
C18:3n6	nd	nd	0.22 ± 0.03	0.35 ± 0.02
C18:3n3	1.25 ± 0.02	0.20 ± 0.03	nd	nd
C20:0	0.17 ± 0.01	0.44 ± 0.51	0.64 ± 0.12	0.79 ± 0.02
C20:1	nd	0.34 ± 0.22	nd	nd
C20:2	0.60 ± 0.10	nd	0.02 ± 0.01	0.03 ± 0.01
C22:0	0.01 ± 0.01	0.26 ± 0.02	0.36 ± 0.13	0.29 ± 0.01
C20:3n6	0.07 ± 0.01	nd	nd	nd
C20:4n6	0.32 ± 0.01	nd	nd	nd
C23:0	nd	nd	0.01 ± 0.01	0.01 ± 0.01
C24:0	nd	0.02 ± 0.01	0.10 ± 0.01	0.09 ± 0.01
C22:6n3	nd	nd	0.02 ± 0.01	0.02 ± 0.01
Σ SFA	39.40 ^b^ ± 0.19	42.97 ^a^ ± 0.28	40.35 ^b^ ± 0.31	42.43 ^a^ ± 0.23
Σ MUFA	44.47 ^a^ ± 0.17	37.01 ^b^ ± 0.04	21.86 ^d^ ± 0.10	29.78 ^c^ ± 0.04
Σ PUFA	16.13 ^d^ ± 0.03	20.03 ^c^ ± 0.10	37.79 ^a^ ± 0.08	27.79 ^b^ ± 0.02
Σ UFA	60.60 ^a^ ± 0.19	57.04 ^d^ ± 0.13	59.65 ^b^ ± 0.03	57.57 ^c^ ± 0.06
UFA/SFA	1.54 ^a^ ± 0.01	1.33 ^c^ ± 0.01	1.48 ^b^ ± 0.01	1.36 ^c^ ± 0.01
MUFA/SFA	1.13 ^a^ ± 0.01	0.86 ^b^ ± 0.01	0.54 ^d^ ± 0.01	0.70 ^c^ ± 0.01
PUFA/SFA	0.41 ^d^ ± 0.01	0.47 ^c^ ± 0.01	0.94 ^a^ ± 0.01	0.65 ^b^ ± 0.01

SFA = saturated fatty acids; MUFA = mono-unsaturated fatty acids; PUFA = poly-unsaturated fatty acids; UFA = unsaturated fatty acids; nd = not detected. a–d: Different letters in the same row correspond to statistically significant differences (*p* ≤ 0.05).

**Table 3 foods-10-01393-t003:** Triacylglycerol (TAG) compositions and cholesterol contents (mean ±SD) of the fat blends L, P, V1, and V2 (for identification of the samples see Table 1).

TAG (%)	L	P	V1	V2
C24-C42	11.8 ± 0.3	0.3 ± 0.1	20.1 ± 0.2	20.5 ± 0.2
C44	0.3 ± 0.1	0.2 ± 0.1	7.0 ± 0.2	5.1 ± 0.1
C46	0.8 ± 0.1	1.0 ± 0.1	6.0 ± 0.2	5.0 ± 0.1
C48	4.8 ± 0.3	7.6 ± 0.1	22.4 ± 1.7	13.0 ± 0.1
C50	19.5 ± 0.6	32.0 ± 0.2	nd	5.6 ± 0.1
C52	44.1 + 6±1.2	34.4 ± 0.4	10.4 ± 0.1	11.3 ± 0.1
C54	18.7 ± 0.3	23.5 ± 0.2	32.8 ± 1.1	38.8 ± 0.9
Others	nd	1.0 ± 0.1	1.3 ± 0.1	0.7 ± 0.2
Cholesterol (%)	0.11 ± 0.3	nd	nd	nd

nd = not detected.

**Table 4 foods-10-01393-t004:** Free fatty acids (FFA), peroxide values (PV), and total polar compounds (TPC) of the SRNs produced with the different fat blends, L, P, V1 and V2.

Samples	FFA (g oleic acid/100 g oil)	PV (meq O_2_/kg oil)	TPC (%)
Raw SRN-L	0.58 ± 0.1 ^c^	3.53 ± 0.1 ^e^	8.84 ± 0.2 ^fg^
Baked SRN-L	0.72 ± 0.1 ^b^	15.75 ± 0.3 ^a^	14.55 ± 0.1 ^c^
Raw SRN-P	0.20 ± 0.1 ^d^	3.23 ± 0.1 ^e^	9.17 ± 0.1 ^ef^
Baked SRN-P	0.35 ± 0.1 ^e^	9.13 ± 0.1 ^b^	15.22 ± 0.1 ^b^
Raw SRN-V1	0.58 ± 0.1 ^c^	3.03 ± 0.1 ^e^	9.65 ± 0.1 ^e^
Baked SRN-V1	0.83 ± 0.1 ^a^	7.25 ± 0.2 ^c^	16.44 ± 0.1 ^a^
Raw SRN-V2	0.11 ± 0.1 ^e^	2.88 ± 0.1 ^f^	9.19 ± 0.1 ^ef^
Baked SRN-V2	0.18 ± 0.1 ^e^	5.35 ± 0.16 ^d^	13.39 ± 0.1 ^d^

a–g: Different letters in the same column correspond to statistically significant differences (*p* ≤ 0.05).

**Table 5 foods-10-01393-t005:** Fatty acid compositions (mean ± SD) of the raw SRNs produced with the different fat blends, L, P, V1, and V2.

Fatty Acids (%)	Raw SRN-L	Raw SRN-P	Raw SRN-V1	Raw SRN-V2
C4:0	0.23 ± 0.02	0.65 ± 0.04	0.14 ± 0.02	0.74 ± 0.13
C6:0	0.16 ± 0.02	0.33 ± 0.02	0.13 ± 0.01	0.42 ± 0.06
C8:0	0.10 ± 0.02	0.25 ± 0.02	0.73 ± 0.08	0.96 ± 0.16
C10:0	0.28 ± 0.02	0.50 ± 0.02	0.66 ± 0.04	1.52 ± 0.11
C12:0	0.32 ± 0.01	0.33 ± 0.07	4.21 ± 0.29	4.45 ± 0.38
C14:0	2.31 ± 0.07	1.75 ± 0.13	2.68 ± 0.13	3.55 ± 0.23
C14:1	0.11 ± 0.01	0.23 ± 0.01	nd	0.26 ± 0.01
C15:0	0.15 ± 0.01	0.23 ± 0.01	nd	0.22 ± 0.02
C16:0	28.21 ± 1.01	40.88 ± 2.30	17.06 ± 0.55	12.59 ± 0.80
C16:1	2.63 ± 0.08	0.68 ± 0.04	1.27 ± 0.05	0.69 ± 0.04
C17:0	0.35 ± 0.02	0.15 ± 0.01	0.19 ± 0.01	0.16 ± 0.01
C17:1	0.37 ± 0.01	nd	0.16 ± 0.01	nd
C18:0	12.54 ± 0.63	3.46 ± 0.59	13.78 ± 0.53	19.85 ± 1.11
C18:1n9t	0.42 ± 0.02	0.26 ± 0.05	0.23 ± 0.01	0.25 ± 0.03
C18:1n9c	34.85 ± 0.03	32.77 ± 1.43	31.07 ± 0.36	29.64 ± 0.44
C18:2n6c	13.79 ± 0.24	17.37 ± 1.53	25.54 ± 0.13	23.68 ± 0.19
C18:3n6	0.22 ± 0.02	nd	0.39 ± 0.02	nd
C18:3n3	0.80 ± 0.02	0.16 ± 0.06	0.52 ± 0.01	0.44 ± 0.02
C21:0	1.00 ± 0.11	nd	0.49 ± 0.02	0.30 ± 0.04
C22:0	1.16 ± 0.13	nd	0.49 ± 0.04	nd
C22:1n9	nd	nd	0.10 ± 0.01	nd
C20:4n6	nd	nd	0.16 ± 0.01	0.28 ± 0.03
Σ SFA	46.81 ^a^ ± 0.28	48.53 ^a^ ± 2.04	40.56 ^b^ ± 0.52	44.76 ^ab^ ± 0.73
Σ MUFA	38.38 ^a^ ± 0.06	33.94 ^b^ ± 1.43	32.83 ^b^ ± 0.33	30.284 ^b^ ± 0.49
Σ PUFA	14.81 ^b^ ± 0.31	17.53 ^b^ ± 1.61	26.61 ^a^ ± 0.19	24.40 ^a^ ± 0.24
UFA/SFA	1.14 ^a^ ± 0.00	1.06 ^a^ ± 0.11	1.47 ^a^ ± 0.03	1.23 ^a^ ± 0.04
TFA	0.42 ^a^ ± 0.03	0.26 ^ab^ ± 0.03	0.23 ^b^ ± 0.01	0.25 ^b^ ± 0.02
Σ omega 3	0.80 ^a^ ± 0.14	0.16 ^c^ ± 0.06	0.52 ^ab^ ± 0.02	0.44 ^bc^ ± 0.01
Σ omega 6	14.01 ^c^ ± 0.33	17.37 ^b^ ± 1.54	26.09 ^a^ ± 0.17	23.96 ^a^ ± 0.22

SFA = saturated fatty acids; MUFA = mono-unsaturated fatty acids; PUFA = poly-unsaturated fatty acids; UFA = unsaturated fatty acids; TFA = *tran*s fatty acids; nd = not detected. a–c: Different letters in the same row correspond to statistically significant differences (*p* ≤ 0.05).

## Data Availability

Research data are not shared.

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
