# Peer review of "Sfogliatella Riccia Napoletana”: Realization of a Lard-Free and Palm Oil-Free Pastry"

_foods, 2021, doi:10.3390/foods10061393_

Round 1

Reviewer 1 Report

The following revisions are recommended:

  • Abstract/ The abstract does not explain the meaning of the paper. The suggestion to the authors is to be specific and precise in the abstract and to state the process parameters that were varied during the experiment. It is necessary to provide insight into the analyzes performed, as well as specific results. Also, it is necessary to write in one sentence a specific conclusion of the conducted research.
  • Keywords are not the best choice. They do not fully cover the topic of the manuscript.
  • Equations: All mathematical equations and the statement/explanation of their variables in the text should be written using an acceptable tool (e.g., MS equation).
  • The presentation of the results was not done correctly. The results presented in the manuscript are interesting, but some of the figures and tables are difficult to interpret. This part of the manuscript is where most of the corrections need to be made. Each result (in the figure or table) must be presented in a clear and understandable way (results are presented in tables/figures, which contain a number of abbreviations, therefore, the interpretation of the results is very difficult). For example, the title of Figure 2 and Table 3 should also include explanations of the abbreviations used. Table 2 certainly needs to be better structured; the results presented in it are difficult to follow now. Etc.
  • Conclusions: Given the scope of the results presented, it is suggested that this section be improved. Focus more on how your research contributed to gaps in knowledge; describe research limitations for future research and restate your major findings; add the scientific and practical significance of the chosen method.

Author Response

Our response in the attached file

Reviewer 2 Report

The manuscript ““Sfogliatella riccia napoletana”: realization of a lard-free and palm oil-free pastry” is generally very well written and contains data of some relevance for a general readers as well as of high relevance for specialists in the topic. Although the subject of the paper could be of interest for the readers of the journal, the paper needs some corrections.

  • Materials and Methods: Why were these oils selected for research and how were the blends composed? Has preliminary research been carried out?
  • Page 3, line: 131 - Was the peroxide value analyzed for lard only?
  • Page 4, lines: 157 and 174 - I am asking for unification in the text: “/min” or “min-1
  • I miss the reference of the results to the literature. Are there any other studies on this type of product? In my opinion, the “Results and Discussion” should be supplemented with a few literature items.
  • In my opinion, it is worth including photos of the obtained products (SRN).

Author Response

Our response in the attached file

Reviewer 3 Report

COMMENTS : FOODS  1246991

Title : “Sfogliatella riccia napoletana”: realization of a lard-free and 2 palm oil-free pastry.

Raffaele Romano, Alessandra Aiello, Lucia De Luca, Alessandro Acunzo, Immacolata Montefusco, Fabiana Pizzolongo.

In this article, the authors looked for an alternative to the use of lard or palm oil for the production of  the traditional "'Sfogliatella riccia napoletana" (SRN). Instead,  they used a mixture of sunflower oil, coconut oil and Shea butter. They studied a number of parameters of these preparations for different fat blends used : lard (L), palm oil (L), and two of the above mentioned mixtures (V1 and V2). They analyzed FFA (free fatty acids), PV(peroxide value), FA composition and % TAG (and carbone number) of fat blends used. They also analyzed cholesterol content, moisture content, water activity, FFA, PV and polar compounds (TPC) of SRN.

General comments :

The presentation of this work is clear. The reviewer really appreciated that the material and method section was very detailed (especially the GC analysis procedures). The experimental procedures seem well mastered.

Major points:

-Concerning the statistical section and/or figures : I did not see anywhere the number of points of the analyses (n). It should be indicated either in the statistical section or in the figures.

-In fig 5, the decrease of C18:0 is not commented. Why this saturated fat is much altered than the C18:2n6c after baking ?

-Line 456-457 : why do you claim that “TFA were lower in vegetable fats without palm oil” ?

Because in table values for TFA for P, V1 and V2 are not significantly différent.

Minor points :

-I don't quite understand the meaning of the double lines (vertical or horizontal) at lines 282-283. Are they necessary?

-In table 4,  I feel that it should be indicated baked SRN-L (rather than SRN-L alone). Ditto for the ones below.

Author Response

Our response in the attached file

Round 2

Reviewer 2 Report

Thank you for considering my suggestions.